# Non-Insulin Novel Antidiabetic Drugs Mechanisms in the Pathogenesis of COVID-19

**DOI:** 10.3390/biomedicines10102624

**Published:** 2022-10-19

**Authors:** Teodor Salmen, Valeria-Anca Pietroșel, Bianca-Margareta Mihai, Ioana Cristina Bica, Claudiu Teodorescu, Horia Păunescu, Oana Andreia Coman, Doina-Andrada Mihai, Anca Pantea Stoian

**Affiliations:** 1Doctoral School, “Carol Davila” University of Medicine and Pharmacy, 020021 Bucharest, Romania; 2Department of Diabetes, Nutrition and Metabolic Diseases, “Prof. Dr N.C.Paulescu” National Institute of Diabetes, Nutrition and Metabolic Diseases, 030167 Bucharest, Romania; 3Department of Pharmacology and Pharmacotherapy, Faculty of Medicine, “Carol Davila” University of Medicine and Pharmacy, 020021 Bucharest, Romania; 4Department of Diabetes, Nutrition and Metabolic Diseases, Carol Davila University of Medicine and Pharmacy, Bld. Eroii Sanitari No. 8, 050471 Bucharest, Romania

**Keywords:** dipeptidyl peptidase-4 inhibitors, glucagon-like peptide-1 receptor agonist, sodium-glucose co-transporter-2 inhibitors, COVID-19, pathogenesis

## Abstract

The present study aimed to analyse the published data and to realize an update about the use and pathogenesis of the novel antidiabetic drugs, respectively, dipeptidyl peptidase-4 inhibitors (DPP-4i), glucagon-like peptide-1 receptor agonists (GLP-1 Ra), and sodium-glucose co-transporter-2 inhibitors (SGLT-2i), in patients with type 2 diabetes mellitus (T2DM) and coronavirus disease (COVID-19). Literature research in the PubMed and Web of Science database was performed in order to identify relevant published clinical trials and meta-analyses that include information about the treatment with novel antidiabetic agents in patients with T2DM and COVID-19. A total of seven articles were included, and their primary and secondary outcomes were reported and analysed. DPP-4i has mixed results on mortality in T2DM patients with COVID-19 but with an overall slightly favourable or neutral effect, whereas GLP-1 Ra seems to have a rather beneficial impact, while SGLT-2i may be useful in acute illness. Even if there are limited data, they seem to have favourable efficacy and safety profiles. The available evidence is heterogenous and insufficient to evaluate if the benefits of non-insulin novel antidiabetic drugs in COVID-19 treatment are due to the improvement of glycaemic control or to their intrinsic anti-inflammatory effects but highlights their beneficial effects in the pathogenesis and evolution of the disease.

## 1. Introduction

Type 2 diabetes mellitus (T2DM) represents a pandemic non-communicable chronic disorder, leading to a high burden for the healthcare systems worldwide due to the development of complications and alteration of the quality of the patient’s life [1].

Since its identification in late 2019, the severe acute respiratory syndrome coronavirus (SARS-CoV-2) spread rapidly as an airborne pathogen, leading to coronavirus disease (COVID-19), which, since March 2020, was declared a pandemic by the World Health Organisation [2,3]. Moreover, the restrictions imposed by the need to control the pandemic of COVID-19 impacted the addressability of healthcare services for diagnosis, monitoring, and treatment of chronic disorders [4,5].

SARS-CoV-2 infects humans through an airborne pathway represented mainly by air droplets [2,6]. The SARS-CoV-2 spike (S) protein binds to Angiotensin-Converting Enzyme 2 (ACE2), which is considered to be the main host cell receptor and initiates the process of replication. Therefore, this enzyme is widely present in human organ tissues [6,7]. Nonetheless, the process of virus adhesion, the underlying receptors, and the mechanisms of triggering and perpetuating the immune response are highly complex [2,8].

A significant influence on the severity, morbidity, and mortality of COVID-19 seems to have originated in factors such as advanced age, male gender, inflammatory biological status, elevated levels of inflammatory cytokines, or comorbidities that predispose to immunosuppression or that lead to organ damage [9], such as obesity, cardiovascular diseases (CVD), high blood pressure, diabetes mellitus (DM), chronic obstructive pulmonary disease, chronic liver disease, and malignancies [2,10,11]. In addition, increased levels of angiotensin-converting enzyme 2 and transmembrane protease serine two are also cited as predisposing factors [2,10].

Association of T2DM to COVID-19 alters the patient’s prognosis, predisposing to lower resistance to infection, thus resulting in more severe forms of COVID-19 needed for intensive care unit (ICU) admission and, overall, a poorer outcome [8,12]. Furthermore, the viral infection and consequent inflammatory reaction induce metabolic distress and aggravate hyperglycaemia in patients with DM, even resulting in acute complications. Moreover, hyperglycaemia alone can alter the immune response and may favour hypoxia and the perpetuation of the initial lesion [13].

The need to face these two pandemics simultaneously raised interest in analysing the complex interaction between the use of antidiabetic agents and COVID-19, beginning from the molecular mechanisms involved in the virus adhesion, in the host’s cell’s inflammatory response, in the secondary organ damage and leading to treatment strategies and clinical outcomes [14,15,16].

The novel non-insulin antidiabetic drugs are represented by the class of dipeptidyl peptidase-4 inhibitors (DPP-4i), glucagon-like peptide-1 receptor agonists (GLP-1 Ra), and sodium-glucose co-transporter-2 inhibitors (SGLT-2i). They are reported to be used in the last decade, with good rates of efficacy and in the condition of safety [17].

Literature data about the use of novel antidiabetic drugs in patients with COVID-19, although constantly increasing, are reported from various studies, primarily observational, and with heterogeneous inclusion criteria and patient populations [13,14]. As for DPP-4i, there are mixed results regarding mortality and the underlying molecular interplay between the DPP-4 enzyme and the SARS-CoV-2 virus [15,16,18]. On the other hand, some studies raise awareness for the use of GLP-1 Ra and SGLT-2i in some cases of patients with T2DM that develop COVID-19 [12].

The present study aims to update the use of novel antidiabetic molecules (DPP4-i, GLP-1 Ra, SGLT2-i) in T2DM and COVID-19 patients by highlighting primary and secondary outcomes from clinical studies and underlying possible molecular mechanisms.

## 2. Materials and Methods

We performed a systematic review following the CRD4 Prospero Protocol. In addition, we conducted literature research in the PubMed and Web of Science databases, including original articles, respectively, randomized control trials, and observational studies, from 1 January 2020, up to 20 June 2022, in order to identify relevant full-text, in English, published clinical trials that included information about the treatment with novel antidiabetic agents in patients (both in- and out-patients) with T2DM and COVID-19.

We searched using the following keywords: (glucagon-like peptide-1 receptor agonists OR sodium-glucose cotransporter-2 inhibitors OR dipeptidyl peptidase-4 inhibitors) AND (COVID-19) AND (type 2 diabetes mellitus OR diabetes OR diabetes mellitus OR type 2 diabetes). As a result, we identified 59 relevant articles written in English (2 on PubMed and 57 on Web of Science) and excluded one article because it was a duplicate; furthermore, two researchers screened the titles and abstracts in order to identify relevant articles, and if any disagreements occurred in the selection process, they were settled down by a third reviewer. As a result, twelve articles were selected, and after the bias assessment using the Newcastle–Ottawa Scale [19], seven articles were included, as shown in Figure 1.

We also performed a manual search of the references to identify other potentially useful articles missed by our search strategy. Finally, in the Discussion section, we completed this information to provide a more comprehensive picture of the available data from the literature.

The present research evaluated the data reported in each of the articles analysed and described the results as main outcomes if they either were the study’s preliminary results or were presented as the study’s main outcomes. The secondary outcomes were either derived from a subgroup analysis of the study or were presented as secondary study outcomes.

## 3. Results

The results of our research are synthesized in Table 1, which contains data about the authors (in the “Author” column), date of publishing (in the “Date of publishing” column), the type of study (in “Study type” column), the total number of patients enrolled (in “Number of patients” column), the main outcomes reported by each article in its result section (in “Main outcome” column), and the secondary outcomes reported by each article in its result section (in “Secondary outcomes” column).

Kahkoska et al. included 12,446 individuals from the U.S. with a positive SARS-CoV-2 PCR test and at least one ambulatory prescription of either GLP1-Ra, SGLT2-i, or DPP-4i in the 24 months preceding the SARS-CoV-2 PCR test. This observational, multicentre study showed that patients with premorbid GLP1-RA or SGLT2i prescribing had lower mortality rates and other adverse clinical outcomes (total mortality, emergency room visits, hospitalization, and mechanical ventilation within 14 days of COVID-19 diagnosis) as compared with DPP4i prescribing, as seen in Table 1. As authors note, the study focuses on comparisons of drug prescribing rather than by drug initiation and several biases should be considered, particularly that patients treated with DPP-4i were older and associated with more comorbidities and had lower use of insulin [20].

Elibol et al. performed a cross-sectional study on 432 patients with T2DM and at least a 3 month-prescription of an antihyperglycemic drug, diagnosed with COVID-19. The patients were stratified into two groups, patients with and without pneumonia according to thoracic CT scan findings, and mortality rates and mortality-related parameters were compared between survivors and non-survivors. As regarding the antidiabetic molecules of interest, pneumonia development was associated with DPP-4i use (*p* = 0.01). SGLT-2i use was not included in the regression analysis, because only patients in the pneumonia group had SGLT-2i use. There was no significant difference between class representants. Additionally, there was no association between DPP-4i and SGLT-2i use with 75-days mortality, as seen in Table 1. Limitations of the study consisted of incomplete documentation about history of COVID-19 exposure and laboratory tests, in particular, A1c values, thus the analysis regarding association between degree of glycaemic control and severity and mortality of COVID-19 was limited [21].

In a multinational retrospective cohort study, Nyland et al. examined usage of GLP1-Ra, DPP-4i, pioglitazone prescribed with at least 6 months preceding the first record of COVID-19, in relation to the incidence of hospital admissions, respiratory complications, and mortality within 28 days after a COVID-19 diagnosis. Overall, 28-day mortality was reduced for both GLP1-Ra and DPP-4i users. After matching for age, sex, race, ethnicity, BMI, and significant comorbidities, the use of GLP1-Ra was associated with significant reductions in hospital admissions, respiratory complications, and incidence of mortality. The use of DPP-4i was associated with a reduction in respiratory complications and continued use of DPP-4i after hospitalization was associated with a decrease in mortality compared with those who discontinued use. Furthermore, in the control group (patients not receiving any of the molecules above mentioned) A1c levels were higher, and it may be implied that factors other than glycaemic control may be responsible for the protective effects of GLP1-Ra and DPP-4i in patients with T2DM and COVID-19. The study had common limitations such as incomplete data on duration of use of medications before hospital admission, dosage, treatments received after admission, and the retrospective nature [22].

In a secondary analysis of the CORONADO study performed by Roussel et al. on 2449 patients with T2DM hospitalized for COVID-19 in 68 French centres, there were similar rates of occurrence of a primary composite endpoint, combining tracheal intubation for mechanical ventilation and death within seven days of admission in users and nonusers of DPP-4i (prior usage to hospitalization). The pattern was similar when outcomes were reassessed at day 28, except for a trend of a non-significant reduction in mortality in DPP-4i users (Table 1). However, DPP-4i users were more frequently under renin-angiotensin-aldosterone system blockers and, upon admission, appeared to have a slightly more severe form of infection, with higher plasma glucose and C-reactive protein (CRP) concentrations. The study also offered information on how DPP-4 inhibitors were handled during hospitalization, with the majority of them (81%) remaining on treatment, including those who had a transitory suspension or a change in dosage, while 19% had stopped treatment, although authors admit that these extensive data on the use of antidiabetic drugs may not be reliably collected. This was the largest cohort analysed to date to test the safety of this class of drugs during the course of the SARS-CoV-2 pandemic [23].

In a retrospective study including 920 patients, Israelsen et al. showed that neither current use (defined as redeemed prescriptions within 90 days before testing positive for SARS-CoV-2) of GLP-1 RA nor DPP-4i were associated with improved outcomes of individuals with diabetes infected with SARS-CoV-2 when compared with SGLT-2i use. The primary outcome was death within 30 days after a positive SARS-CoV-2 test, while secondary outcomes included hospital admission, intensive care unit (ICU) admission, and mechanical ventilation. Patients were followed from the date of positive test for SARS-CoV-2 until death, migration, or end of follow-up (30 days). In weighted analyses, GLP-1 RA users had similar 30-day mortality to SGLT-2i users (3.3% vs. 3.7%) that both were lower than for DPP-4i users. Risks of hospital admission, ICU admission, and mechanical ventilation were overall similar in GLP-1 RA and DPP-4i users compared with SGLT-2i users. Risks of hospital admission and mechanical ventilation were increased across all treatment groups with adjusted RRs between 1.22 and 2.22, while adjusted RR for ICU admission was close to 1.0 for GLP-1 RA users and 1.30 for DPP-4i users. Less than 30% of individuals were hospitalized. Limitations involve the retrospective character, limited sample size and low statistical precision, and association of DPP-4i use with older age and higher Charlson’s Comorbidity Index (CCI) scores [24].

In a multicentre, retrospective analysis, including 2563 patients with T2DM who were hospitalized due to COVID-19 at 16 hospitals in Hubei Province, China, Zhou et al. found that after propensity matching, there was no significant association between in-hospital DPP-4i use and 28-d all-cause mortality (adjusted hazard ratio = 0.44, 95% CI: 0.09–2.11, *p* = 0.31), and the incidences and risks of secondary outcomes, including septic shock, acute respiratory distress syndrome, or acute organ (kidney, liver, and cardiac) injuries, were also comparable between the DPP-4i and non-DPP-4i groups. Glucose control was mostly equivalent between groups. Furthermore, the authors did not observe substantial side effects such as uncontrolled glycemia or acidosis due to DPP-4i usage. Similar to other studies, limitations involve the retrospective character and small sample size. Furthermore, the results may not be applicable to out-patients, as all subjects enrolled in this study were hospitalized [25].

Solerte et al., in a multicentre, case-control, retrospective study on 338 patients with T2DM hospitalized for COVID-19, showed that sitagliptin treatment at the time of hospitalization was associated with reduced mortality and improved clinical outcomes as compared with standard-of-care treatment (insulin) and with an improvement in clinical outcomes, decreased risk for the need for mechanical ventilation, reduced risk for the need for I.C.U. admission, and with a greater number of hospital discharges compared with patients receiving standard of care, respectively (Table 1). However, considering the observational nature of this observation, the impact of confounders on the outcomes should be considered in patients treated with DPP-4i compared to patients treated with insulin [26].

## 4. Discussion

### 4.1. DPP-4i

DPP-4, originally known as CD26, a serine protease that can be found in a membrane-bound and, respectively, in a soluble form that maintains its enzymatic activity [27], is expressed almost ubiquitously in the human body, in the intestine, spleen, pancreas, liver, lung, bone marrow, and kidney [28,29], as well as on the surface of different cell types, primarily endothelial, epithelial, and immune cells [16,28,29]. It cleaves a wide variety of substrates, including the incretin hormones glucagon-like peptide-1 (GLP-1) and gastric inhibitory peptide (GIP), cytokines, and growth factors. It also acts as a binding protein and a ligand of extracellular factors [18,30,31,32], hence the pleiotropic effects of DPP-4 enzyme [32].

DPP-4 and DPP-4i roles in COVID-19 have gathered increasing interest. Although it was speculated that DPP-4 could be a functional receptor for SARS-CoV-2 that could facilitate viral entry, recent studies have demonstrated that SARS-CoV-2 spike protein does not interact with human membrane-bound DPP-4 (CD26) [14,15,33]. DPP-4i may have beneficial effects in patients with T2DM and COVID-19 through mechanisms, such as the antihyperglycemic effect with subsequent positive implications on inflammation, endothelial injury, and viral proliferation [13,32,34], but also through direct anti-inflammatory, immuno-modulatory, and antifibrotic effects [18].

DPP-4i, also called gliptins, are known as oral antidiabetic drugs. In the last decade of use, they became widely available due to their efficacy and good safety profile with minimal risk of hypoglycaemia [28,32]. The DPP-4-mediated inactivation and cleavage of incretin hormones released after meal ingestion, and DPP-4i suppress glucagon release while enhancing pancreatic insulin secretion [35] proportionally to the glucose uptake, thus regulating postprandial blood glucose levels [28].

It is hypothesized that DPP-4 levels are dysregulated in patients with T2DM, which may have a negative vascular impact, hence leading to increased severity of COVID-19 [27]. Furthermore, DPP-4 has a higher distribution in visceral fat tissue, contributing to insulin resistance and adipocyte inflammation. Since obesity is also frequently associated with poorer outcomes in COVID-19 patients, DPP-4 inhibition may bring some advantages [18]. Although cardiovascular protective mechanisms of DPP-4i have been thoroughly debated, large cardiovascular outcome trials showed neutrality for DPP-4i in reducing major cardiovascular events except for saxagliptin, which increased the risk for heart failure hospitalization [35,36]. Nonetheless, due to the neutral effect on body weight and optimal safety, with minimal risk for hypoglycaemia and other adverse reactions, gliptins are recommended as add-on therapy to improve glycaemic control and can be continued in patients with mild and moderate forms of COVID-19 [36,37,38].

DPP-4 is widely expressed in many types of immune cells, including CD4(+) and CD8(+) T cells, as well as B cells, N.K. cells, dendritic cells, and macrophages, and modulates their functions [15,29]. DPP-4 directly promotes T cell activation and proliferation, activation of the NF-κB signalling pathway leading to overproduction of pro-inflammatory cytokines [16,31,37], or its effects may be triggered after interaction of antigen presenting cells with markers such as CD45, caveolin-1, mannose-6 phosphate receptor, or adenosine deaminase (A.D.A.) [27]. DPP-4 facilitates immune cell migration and diapedesis through its interaction with A.D.A. [27]. Some indirect anti-inflammatory effects of DPP-4i may be mediated by GLP-1-dependent mechanisms as well, since GLP-1 also exerts anti-inflammatory properties [16,30].

DPP-4i use was associated with lower levels of pro-inflammatory mediators such as interleukin-1 (IL-1), interleukin-6 (IL-6), tumour necrosis factor-α (TNF-α), and C-reactive protein (C.R.P.) in several studies [27,31,37]. However, when used in COVID-19 patients in acute settings, mixed results were obtained [35]. Last but not least, although controversial, DPP-4i may prevent the degradation of certain inflammatory factors; a paradoxical pro-inflammatory effect of DPP-4i cannot be excluded [31].

DPP-4i is also considered to improve outcomes in patients with COVID-19 and lung injuries by decreasing the cytokine storm, thus preventing acute respiratory distress syndrome (ARDS), the common cause of COVID-19-related death [16,31,37]. In an experimental model in mice, sitagliptin was observed to suppress the lipopolysaccharide-induced lung injury and thus improve the histological findings by decreasing the release of cytokines, such as TNF-α and IL-6 [37,38]. In addition, potential antifibrotic effects of DPP-4i have also been speculated, as DPP-4 has been found to promote cytokine and chemokine progression and smooth muscle cell proliferation by fibroblasts. Therefore, DPP-4 inhibition may prevent lung fibrosis progression and reduce mechanical complications of COVID-19 [15].

### 4.2. SGLT-2i

In the case of patients with T2DM, SGLT-2i is a class of antidiabetic drugs that acts at the renal level, inhibiting glucose reabsorption, thus decreasing its level, so they are beneficial, especially when concomitant comorbidities such as atherosclerotic CVD or renal involvement are present. SGLT-2i have multiple and pleiotropic effects—for example, endothelial function improvement that may intervene in limiting the development of thrombo-embolic complications and anti-inflammatory effects that reduce the markers of inflammation such as interleukin 6, ferritin, or C-reactive protein (C.R.P.), and lead to a decrease in the intensity of the cytokine storm [39]. Even if there is no reported inferiority compared to the classes of incretin-based antidiabetic drugs, attention should be paid to the risk of developing diabetic ketoacidosis [39].

When using SGLT-2i in patients with T2DM and COVID-19, we should consider both the class’s positive and negative effects. 

From a positive point of view, data are reported in studies both on mice and on humans regarding its anti-inflammatory effects, such as decreasing the levels of inflammatory cells at the arterial plaques levels, with a limited endothelial dysfunction, independent of the metabolic benefits on glucose and A1c haemoglobin (HbA1c); with a lower expression of mRNA in C reactive protein (C.R.P.) or high sensitivity C.R.P. and monocyte chemoattractant protein 1 (both in mice and human), and IL-6, ferritin, and TNF (only on studies on humans) [40,41,42,43]. Another mechanism involved in improved endothelial function is the reduction of oxidative stress. The metabolic benefits seem not to be limited to the inhibition of glucose reabsorption, with an improvement in the glucose levels, but also extended to a metabolism shift to lipid oxidation and limitation in the glycolysis process that seems to be used by SARS-CoV-2 [44].

On the other hand, the negative aspect refers to an increased risk for ketoacidosis, acute kidney injury caused by dehydration, and uric acid depletion in the context of osmotic diuresis, an aspect especially found in severely ill patients. Therefore, it may be wise that for patients with significantly reduced glomerular filtration rate, even if they are under the strict surveillance of the hydric balance, SGLT-2i should be changed with other classes [40,45].

In patients with COVID-19 and T2DM, there seems to be no difference in 30-days mortality and hospital admission between a previous treatment of T2DM with GLP-1 Ra or DPP4-i compared to SGLT-2i [45].

### 4.3. GLP-1 Ras

GLP-1 Ra is incretin mimetics, with an anti-hyperglycaemic effect based on maintaining pharmacologic levels of GLP-1 that increases insulin secretion in a glucose-dependent manner, decreases glucagon secretion, and delays gastric emptying [45,46]. GLP-1 Ras are potent injectable antidiabetics and have been shown to reduce HbA1c by approximately 0.8–1.6 percentage points [45,46]. Reducing hyperglycaemia linked to immune deficiency and viral proliferation, as well as glycaemic variability, considered a promoter of oxidative stress and consequent inflammation, GLP-1 Ra may represent a valuable treatment option for non-critically ill patients T2DM patients with COVID-19 [46].

Moreover, it is presumed that GLP-1 Ra, by increasing insulin levels and subsequent interference with protein glycosylation, especially that of ACE2, may also impede cellular virus entry [40]. The GLP-1 Ra’s low risk of hypoglycaemia was corroborated with the reduction of catabolism, secondary to glucagon suppression, which was also considered in hospitalized critically ill patients. However, delayed gastric emptying, relatively common in these cases, raised concern about their safety [47].

GLP-1 Ra exerts significant anti-inflammatory properties [46]. For instance, Shao et al., 2020, reported a reduction in plasmatic levels of pro-inflammatory cytokines such as TNF-α, IL-1β, and IL-6, and an increase in adiponectin (which belongs to the anti-inflammatory adipokines) in T2DM patients [48].

Belancic et al., 2021, offered a comparative perspective on the molecular mechanisms involved in the anti-inflammatory properties of GLP-1. Accordingly, binding of GLP-1/GLP-1Ra to the GLP-1R results in the blockage of the protein kinase C or nuclear factor-ĸB (NF-ĸB) activation and subsequent reduction in the expression of NOD-like receptor (NLR) family pyrin domain containing 3 (NLRP3), IL-1β, TNF-α, IL-6, vascular cell adhesion molecule 1, interferon-γ, and monocyte chemoattractant protein-1 [46]. Further description of interference with the NF-ĸB signalling pathways and NLRP3-mediated inflammation is proposed by Banerjee et al., 2021. They also discussed the possible mitochondrial protective and antithrombotic effects of GLP-1 Ra [45]. Other signalling cascades and targets of GLP-1-based drugs assumed to reduce the expression of the pro-inflammatory cytokines were also considered [49].

Multiple studies in rodent models with experimental lung injuries brought evidence that GLP-1 and GLP-1-based drugs attenuate the cytokine storm, thus exerting a beneficial immune-regulatory and pulmonary protective effect [45,46,48,49]. Rogliani et al., 2019, recently reported that treatment with GLP-1 Ra improves lung function regardless of the blood glucose levels in T2DM patients without underlying obstructive pulmonary disorders, suggesting that GLP-1 Ras have a direct effect on lung tissue [50]. Based on their anti-inflammatory effect on the lungs, a beneficial role of GLP-1 Ra in treating pulmonary arterial hypertension (another possible complication of COVID-19) is also hypothesized [51].

Anti-obesogenic properties of long-acting GLP-1 Ra could be of significant value, since obesity is associated with increased COVID-19 susceptibility and severity [45,46,52,53]. In addition, GLP-1 Ras’s beneficial role in patients with T2DM and high risk for CVD, including its anti-atherosclerotic effect, may also play a part in preventing severe forms of COVID-19, but more evidence is needed [14,45,52]. Last but not least, liraglutide/GLP-1 Ra may have beneficial effects on gut microbiota, preventing microbiome dysbiosis and endotoxemia, thus leading to a less severe form of coronavirus disease [46].

The interaction between GLP-1 and ACE2 has raised considerable debate [45,51]. Although ACE2 enables virus entry into host target cells, ACE2 upregulation induced by GLP-1 Ra may, in turn, lead to a paradoxically favourable effect and ameliorate lung injury during COVID-19 [14,45,46,51].

GLP-1 Ra seems to have a rather favourable effect in SARS-CoV-2 infected patients [14,45,51,54] and may represent an appropriate treatment for non-critically ill patients with T2DM [40,52]. However, the use of GLP-1 Ra in COVID-19 treatment is still under debate [12,46,51], and more studies, especially clinical trials, are necessary to bring a more accurate perspective [14,51].

To sum up, several meta-analyses [16,18,31,37,54,55] have assessed the potential benefits of novel antidiabetic molecules when treating T2DM patients with COVID-19. Because COVID-19 is a pathology with a recent outbreak, there are few studies with a small number of enrolled patients, and the vast majority of these studies are retrospective and observational. Another major limitation of the meta-analyses mentioned above was the large heterogeneity across the studies, with several explanations. For instance, in DPP4-i meta-analyses, most of the studies were not designed to directly investigate the role of DPP-4i in patients with T2DM and COVID-19, as Bonora et al., 2021 [31] noted, or there were different settings of DPP-4 administration (e.g., inpatients vs. outpatients, patients initiated to DPP-4i when hospitalized vs. already on DPP-4i treatment when COVID-19 infection was confirmed) as in the meta-analyses conducted by Patoulias et al., 2021 [37], Yang et al., 2021 [16], and Rakhmat et al., 2021 [18]. The duration of treatment and use of other antidiabetics were also limitations. Furthermore, the studies had different conceptualizations and various primary outcomes, and the confounding factors were not well balanced (e.g., inclusion criteria and baseline characteristics).

An open-labelled R.C.T. performed by Abuhasira et al., 2021 [56], using linagliptin, reported there was no difference in time to clinical improvement versus standard of care. The study included a tiny sample and was prematurely discontinued due to the end of the COVID-19 outbreak in Israel. Therefore, it was considered underpowered to detect possible differences in the primary outcome and in mortality. Moreover, the authors mentioned a lower A1c level in the linagliptin group, and the group overall had more comorbidities, which could have influenced the outcomes. A meta-analysis performed by Hariyanto et al., 2021 [54], which included only observational studies on patients already on GLP-1 Ra, reported a rather favourable effect of the medication. However, the authors complained about the lack of information regarding the dosage and the duration of treatment. Furthermore, because of two dominant studies included in the meta-analysis, the authors declared possible confounding of the results. As for SGLT2-i, there are limited data regarding their use in the SARS-CoV2 infection. A very small observational study performed by Dalan et al. [57] found a lower risk of mechanical ventilation in patients receiving SGLT-2i, after adjustment for baseline features and CV status.

DARE-19 (Dapagliflozin in Respiratory Failure in Patients With COVID-19) was a trial of 1250 hospitalized patients with COVID-19 and cardiometabolic risk factors (i.e., hypertension, T2DM, atherosclerotic CVD, heart failure, and chronic kidney disease) who were randomized 1:1 to either placebo or dapagliflozin. Although the findings did not reach statistical significance, the efficacy and safety profiles of the drug were favourable, with fewer participants in the intervention arm developing organ failure or death. Related specifically to DKA, there were only two cases reported in individuals with T2DM, both non-severe, suggesting that SGLT-2i may be used in acute illness, provided close monitoring. However, it should be considered that the trial was short (30 days), had a small sample, and the results should be carefully interpreted, as the improvement in the COVID-19 management and reduction in mortality rates can be confounding factors [58,59,60]. However, efficacy and safety of SGLT-2i use in acute illness, particularly with respect to the development of DKA, are still under debate, and further RCTs and real-world data are needed. There are scarce data on comparisons between molecules from the same class and whether the observed results can be considered a class effect or belong only to particular molecules. Consideration should also be given to the time when the studies were conducted, as recommendations for specific COVID-19 management and treatment were changing at a fast pace from the outbreak of the pandemics. Furthermore, there was no consensus, and different medical strategies were proposed. Therefore, different clinical outcomes results may be influenced by the different COVID-19 treatment strategies (i.e., use of specific antivirals or anti-inflammatory drugs). The same case could apply to the simultaneous use of metformin and A.C.E. inhibitors, as evidence suggested that the use of metformin, another antidiabetic drug, and renin-angiotensin system inhibitor were associated with lower severity and mortality in COVID-19 patients [14,61].

Last but not least, the concomitant standard of care therapy can be a significant confounder of the final results. Insulin remains the treatment of choice during acute illness in hospitalized patients to improve glycaemic control and reduce the risk for ketoacidosis. Adequate titration to reach glycaemic targets and to avoid hypoglycaemia is of utmost importance. Oral antidiabetic medications can be used in mild COVID-19 forms without severe hyperglycaemia as long as they are not contraindicated. DPP-4i are generally well tolerated, and presently, there is no firm evidence for the need to be discontinued. Given the risk of dehydration and diabetic ketoacidosis during respiratory illness, SGLT2-i could be discontinued, and initiation is not recommended. GLP-1 Ra may be continued, provided good hydration and adequate food intake [46,47]. The use of SGLT-2i in acute illness is under hard debate, with most guidelines recommending treatment discontinuation given the risk of dehydration and diabetic ketoacidosis during respiratory illness, although some data suggest good efficacy and safety profiles, but more studies are needed to support that [47,58]. Reinitiation could be recommended after discharge [59,62]. Finally, the implications and possible benefits of novel antidiabetic drugs in long-COVID syndrome should be carefully monitored, and randomized trials are urgently needed, as seen in Figure 2.

## 5. Conclusions

The available evidence is insufficient to evaluate if the benefits of non-insulin novel antidiabetic drugs in COVID-19 treatment are due to the molecule itself or to the improvement of glycaemic control. However, some of these drugs may be beneficial due to their intrinsic anti-inflammatory effects.

Concluding, DPP-4i have mixed results regarding the mortality in T2DM patients with COVID-19 and overall seems to have a slightly favourable or neutral effect, whereas GLP-1 Ra seems to have a rather beneficial impact. SGLT-2i use in acute illness is still debatable, and even if data are limited, they seem to have favourable efficacy and safety profiles.

Frequently encountered limitations of the presented studies were the retrospective, observational design, and the presence of confounding factors (e.g., inclusion criteria and baseline characteristics). More prospective studies are needed to evaluate the real impact of novel antidiabetic drugs on COVID-19 pathogenesis and clinical evolution.

## Figures and Tables

**Figure 1 biomedicines-10-02624-f001:**
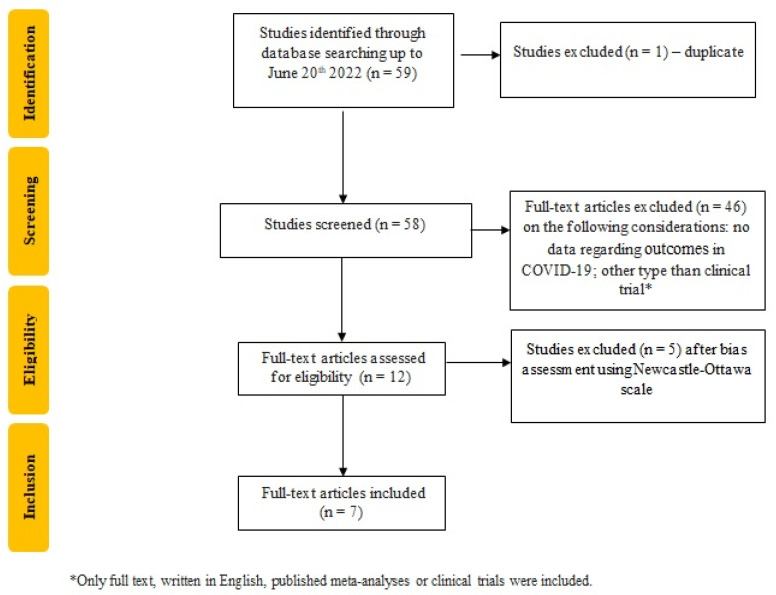
Flowchart of the study selection process. Flowchart of the study selection process according to PRISMA guidelines.

**Figure 2 biomedicines-10-02624-f002:**
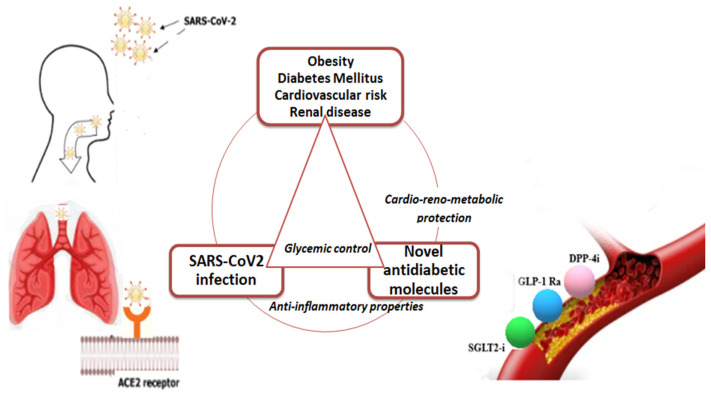
The implications and possible benefits of novel antidiabetic drugs in COVID.

**Table 1 biomedicines-10-02624-t001:** Summary of papers evaluating the association between COVID-19 clinical outcomes and DPP-4i, SGLT-2i, and GLP-1 Ra use.

Secondary Outcome	Main Outcome	Number of Patients	Study Type	Date of Publishing	Author
mortality rate over the observation period was 2.29% for GLP-1 Ra, 2.48% for SGLT2i, and 6.18% for DPP4i. Emergency room visits rate was 28.84% for GLP-1 Ra, 29.3% for SGLT2i, and 36.6% for DPP4i. hospitalizations rate was 21.89% for GLP-1 Ra, 23.22%, for SGLT2i, and 33.38% for DPP4i. The need for mechanical ventilation within 14 days of a positive SARS-CoV-2 test rate was 5.78% for GLP-1 Ra, 6.17% for SGLT2i, and 8.54% for DPP4i.	60-day mortality from a positive SARS-CoV-2 test according to premorbid medication use—2.06% for GLP-1 Ra; 2.32% for SGLT2i; respectively, 5.67% for DPP4i. Both GLP1-RA and SGLT2i use were associated with lower 60-day mortality compared with DPP4i use (OR 0.54 [95% CI 0.37–0.80] and 0.66 [0.50–0.86], respectively).	12,446	Observational study	Epub June 2021	Kahkoska et al. [20]
no association between DPP4i and SGLT2i use with 75-days mortality (Survivor—55.7% vs. non-survivor 61.5% for DPP4-i, *p* = 0.342; Survivor—12.3% vs. non-survivor 15.4% for SGLT2-i, *p* = 0.482)	pneumonia development was associated with DPP-4i use (*p* = 0.01) and SGLT-2i use (*p* = 0.006)	432	cross-sectional study	Epub August 2021	Elibol et al. [21]
Hospital admission -RR 0.53, CI 0.47–0.59, *p* < 0.001, for GLP1-Ra -RR 1.04, CI 0.97–1.11, *p* = 0.3, for DPP4-i Respiratory complications -RR 0.56, CI 0.50–0.63, *p* < 0.001, for GLP1-Ra -RR 0.88, CI 0.82–0.95, *p* = 0.001, for DPP4-i	28-day mortality was reduced: -RR 0.41, CI 0.3–0.55, *p* < 0.001, for GLP1-Ra -RR 1.23, CI 1.05–1.43, *p* = 0.009, for DPP4-i	229,809	retrospective study	Epub September 2021	Nyland et al. [22]
DPP-4i users vs. nonusers of DPP4-i: -Intermittent mechanical Ventilation at 7 days: 19.1% vs. 18.5%, *p* = 0.7169 -Intermittent mechanical Ventilation at 28 days: 20.3% versus 19.2%, *p* = 0.5527 -Death at 7 days: 9.7% versus 11.7%, *p* = 0.2048 -Death at 28 days: 18.1% versus 21.8%, *p* = 0.0561	composite endpoint combining tracheal intubation for mechanical ventilation and death within seven days of admission had similar rates in users and nonusers of DPP-4i (27.7% vs. 28.6%; *p* = 0.68)	2449	Secondary analysis study of the CORONADO study	Epub February 2021	Roussel et al. [23]
ICU admission -GLP-1 Ra versus SGLT-2i RR of 0.89 (0.38–2.07) -DPP-4i versus SGLT-2i RR of 1.73 (0.78–3.85) Mechanical ventilation -GLP-1 Ra versus SGLT-2i RR of 0.95 (0.37–2.46) -DPP-4i versus SGLT-2i RR of 2.41 (0.95–6.13) Hospital admission -GLP-1 Ra versus SGLT-2i RR of 1.17 (0.87–1.57)	30-day mortality after a positive SARS-CoV-2 test -GLP-1 Ra versus SGLT-2i RR of 1.03 (0.45–2.35) -DPP-4i versus SGLT-2i RR of 4.13 (1.85–9.26) -DPP-4i versus SGLT-2i RR of 1.53 (1.14–2.05)	1970	population-based cohort study	Epub February 2021	Israelsen et al. [24]
comparable incidences and risks between the DPP4i and non-DPP4i groups of the secondary outcomes: -occurrences of septic shock (*p* = 0.82); ARDS (*p* = 0.85), acute organ injury (kidney-*p* = 0.98, liver-*p* = 0.25, and cardiac-*p* = 0.28)	28-day all-cause mortality between DPP-4i users and non-DPP4 users: 1.8% vs. 3.3%; OR = 0.58 95% CI: 0.12–2.68, *p* = 0.48)	2563	multicentre retrospective cohort study	November 2020	Zhou et al. [25]
decreased risk for the need for mechanical ventilation as compared with treatment with the standard of care (HR 0.27 [CI 0.11–0.62]; *p* = 0.003) reduced risk for the need for ICU in the sitagliptin-treated group as compared with the standard-of-care group (HR 0.51 [CI 0.27–0.95]; *p* = 0.03) no difference in the requirement of ECMO as compared with treatment with the standard of care (HR 1.15 [CI 0.41–3.17]) overall improvement of clinical score vs. standard of care (60% vs. 38%, *p* = 0.0001)	mortality at 30 days in sitagliptin versus standard of care 18% versus 37%, *p* = 0.0001; in-hospital death (OR 0.37 [CI 0.23–0.62]; *p* = 0.0001) hospital discharge at 30 days in sitagliptin versus standard of care 71% versus 52.66%, *p* = 0.0008; time to clinical endpoint (death/discharge) (hazard ratio [HR] 0.44 [95% CI 0.29–0.66]; *p* = 0.0001	338	multicentre, case-control, retrospective, observational study	Epub September 2020	Solerte et al. [26]

Abbreviation list: RR—risk ratio, CI—confidence interval, OR—odds ratio; HR—hazard ratio; RCT—randomised control trials; DM—diabetes mellitus; DPP-4i—dipeptidyl peptidase-4 inhibitors; GLP-1 Ra—glucagon-like peptide-1 receptor agonists; SGLT-2i—sodium-glucose cotransporter-2 inhibitors; COVID-19—coronavirus disease; ARDS—acute respiratory distress syndrome; ICU—intensive care unit; ECMO—extracorporeal membrane oxygenation.

## Data Availability

Not applicable.

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
