# Peer review of "Non-Insulin Novel Antidiabetic Drugs Mechanisms in the Pathogenesis of COVID-19"

_biomedicines, 2022, doi:10.3390/biomedicines10102624_

Round 1
Reviewer 1 Report
Dear Editor,
I’ve read with great interest the Systematic Review “Non-insulin novel antidiabetic drugs mechanisms in the pathogenesis of COVID-19” by Teodor Salmen et al. However, in the opinion of this reviewer some issues need to be raised.
- Introduction: Recently, some authors have shown that, among risk factors, also chronic liver disease could affect in-hospital mortality in COVID-19 subjects. doi:10.1371/journal.pone.0243700
- A review of the figure is needed. Please improve the quality of the table, maybe a horizontal layout could help.
- The authors write:” use of metformin, another antidiabetic drug, and renin-angiotensin system inhibitor were associated with lower severity and mortality in COVID-19 patients”. However no reference is associated with the description of the use of metformin. Recently, some authors have shown preclinical and clinical evidence favoring the “anti-aging” therapeutic potential of this drug. doi:10.1016/j.diabres.2020.108025
Author Response
Response Letter to reviewers
Dear esteemed reviewers,
Thank you very much for your availability to review our work- the manuscript entitled “Non-
insulin novel antidiabetic drugs mechanisms in the pathogenesis of COVID-19” by
Teodor Salmen et al.
We are grateful for your insights and recommendation, which will improve our article. We
followed your instructions as you indicated, and please find below what we performed.
Reviewer 1
I’ve read with great interest the Systematic Review “Non-insulin novel antidiabetic drugs
mechanisms in the pathogenesis of COVID-19” by Teodor Salmen et al. However, in the
opinion of this reviewer some issues need to be raised.
- Introduction: Recently, some authors have shown that, among risk factors, also chronic liver
disease could affect in-hospital mortality in COVID-19 subjects.
doi:10.1371/journal.pone.0243700
Response: Thank you very much for your observation, we edited it accordingly.
- A review of the figure is needed. Please improve the quality of the table, maybe a horizontal
layout could help.
Response: Thank you for your comment, we changed the table into a portrait one.
- The authors write: ”use of metformin, another antidiabetic drug, and renin-angiotensin
system inhibitor were associated with lower severity and mortality in COVID-19 patients”.
However no reference is associated with the description of the use of metformin. Recently,
some authors have shown preclinical and clinical evidence favoring the “anti-aging”
therapeutic potential of this drug. doi:10.1016/j.diabres.2020.108025
Response: Thank you very much for your comment, but at the end of the paragraph there are
two references for the paragraph and, also, for the phrase cited by you, while the reference
suggested is not about metformin and COVID-19, so we can’t use it.
Reviewer 2 Report
The relationship between anti-diabetic agents and COVID-19 results has gained much attention, mainly due to the plethora of pleiotropic actions of the new glucose-lowering agents. The review by Salmen et al. is well written and rich in information. However, I think there is still room for further improvements.
Specific Points:
1. Abstract: While the authors refer to the effects of DPP4i and GLP-1 RAs on COVID outcome, they mention nothing about SGLT2i.
2. Introduction: 'With reasonable rates of efficacy' GLP-1 RAs not only present 'reasonable' efficacy, but have a higher glucose-lowering potency even compared to insulin.
3. Methods: Please clearly define the inclusion and exclusion criteria for the articles you wished to include in the systematic review. RCTs? Observational studies? Studies in inpatients, outpatients, or both?
4. The results section is very short. The authors should summarize and incorporate the results of this systematic review into the text as well (apart from the table).
5. Included trials must be critically presented. What are the main advantages and limitations of each study?
6. DPP-4i: '...trials showed neutrality for DPP4-i' Saxagliptin has been shown to increase the risk of hospitalization for HF
7. DPP4i: In the Italian study by Solerte et al., sitagliptin treatment at the time of hospitalization was demonstrated to correlate with reduced mortality and improved clinical outcomes. However, considering the observational nature of this observation, the impact of confounders on the outcomes should be considered (patients treated with DPP-4i might be in better physical shape compared to those treated with insulin, for example)
8. SGLT2i: gliflozins cause diabetic ketoacidosis, but not lactic acidosis (which can be caused by metformin). Please, correct.
9. Diabetes and the attack by COVID in different organs share common mechanisms. This is why SGLT2i could alleviate organ damage in the context of COVID infection (please review https://doi.org/10.1007/s11096-021-01256-9 and expand on this aspect)
10. The findings of DARE-19 should be discussed in the SGLT2i section, since this is the only large RCT inthe field.
11. Furthermore, the risk – benefit balance of the administration of SGLT2i in COVID-19 deserves further attention (https://doi.org/10.1016/j.phrs.2021.105872). Furthermore, the potential use of these agents in the hospital setting is relevant (https://doi.org/10.1007/s40265-022-01730-2)
12. COVID has been suggested as a new predisposing CV risk factor (https://doi.org/10.3390/jcdd8100130). Therefore, not only the anti-inflammatory but also the anti-atherosclerotic effects of GLP-1 could be beneficial in the context of COVID-19 and should be discussed in the relevant section.
13. Similar to SGLT2i, the use of GLP-1 in the hospital is relevant and deserves to be mentioned (https://doi.org/10.1002/dmrr.3574).
14. The meta-analysis by Patoulias et al. is not reference 31 but reference 34.
15. Discussion: The discontinuation of SGLT2i has been argued in the context of acute illness as a standard policy. See https://doi.org/10.1111/dom.14805.
Author Response
Response Letter to reviewers
Dear esteemed reviewers,
Thank you very much for your availability to review our work- the manuscript entitled “Non-
insulin novel antidiabetic drugs mechanisms in the pathogenesis of COVID-19” by
Teodor Salmen et al.
We are grateful for your insights and recommendation, which will improve our article. We
followed your instructions as you indicated, and please find below what we performed.
Reviewer 2
The relationship between anti-diabetic agents and COVID-19 results has gained much
attention, mainly due to the plethora of pleiotropic actions of the new glucose-lowering
agents. The review by Salmen et al. is well written and rich in information. However, I think
there is still room for further improvements.
Specific Points:
1. Abstract: While the authors refer to the effects of DPP4i and GLP-1 RAs on COVID
outcome, they mention nothing about SGLT2i.
Response: Thank you for your suggestion, we edited accordingly.
2. Introduction: 'With reasonable rates of efficacy' GLP-1 RAs not only present
'reasonable' efficacy, but have a higher glucose-lowering potency even compared to insulin.
Response: Thank you for your suggestion, we have updated the paragraph starting with “The
novel non-insulin antidiabetic drugs are represented by the class ”, according to the aim of the
study to describe all the three novel antidiabetic drugs classes.
3. Methods: Please clearly define the inclusion and exclusion criteria for the articles you
wished to include in the systematic review. RCTs? Observational studies? Studies in
inpatients, outpatients, or both?
Response: Thank you for your suggestion, we updated accordingly.
4. The results section is very short. The authors should summarize and incorporate the
results of this systematic review into the text as well (apart from the table).
Response: Thank you for your suggestion, we developed the results part.
5. Included trials must be critically presented. What are the main advantages and
limitations of each study?
Response: Thank you for your suggestion, we added the data critically .
6. DPP-4i: '...trials showed neutrality for DPP4-i' Saxagliptin has been shown to increase
the risk of hospitalization for HF
Response: Thank you for your suggestion, we have updated in text.
7. DPP4i: In the Italian study by Solerte et al., sitagliptin treatment at the time of
hospitalization was demonstrated to correlate with reduced mortality and improved clinical
outcomes. However, considering the observational nature of this observation, the impact of
confounders on the outcomes should be considered (patients treated with DPP-4i might be in
better physical shape compared to those treated with insulin, for example)
Response: Thank you for your suggestion, we edited accordingly.
8. SGLT2i: gliflozins cause diabetic ketoacidosis, but not lactic acidosis (which can be
caused by metformin). Please, correct.
Thank you for your suggestion, we corrected.
9. Diabetes and the attack by COVID in different organs share common mechanisms. This
is why SGLT2i could alleviate organ damage in the context of COVID infection (please
review https://doi.org/10.1007/s11096-021-01256-9 and expand on this aspect)
Response: Thank you for your suggestion, we did accordingly
10. The findings of DARE-19 should be discussed in the SGLT2i section, since this is the
only large RCT in the field.
Response: Thank you for your suggestion, we updated the discussions section.
11. Furthermore, the risk – benefit balance of the administration of SGLT2i in COVID-19
deserves further attention (https://doi.org/10.1016/j.phrs.2021.105872). Furthermore, the
potential use of these agents in the hospital setting is relevant
(https://doi.org/10.1007/s40265-022-01730-2)
Response: Thank you for your suggestion, we did accordingly
12. COVID has been suggested as a new predisposing CV risk factor
(https://doi.org/10.3390/jcdd8100130). Therefore, not only the anti-inflammatory but also the
anti-atherosclerotic effects of GLP-1 could be beneficial in the context of COVID-19 and
should be discussed in the relevant section.
Response: Thank you for your comment, we edited the paragraph “Anti‐obesogenic
properties of long‐acting GLP‐1 Ra” as you recommended.
13. Similar to SGLT2i, the use of GLP-1 in the hospital is relevant and deserves to be
mentioned (https://doi.org/10.1002/dmrr.3574).
Thank you for your comment, please re-check the edited paragraph starting with “Moreover,
it is presumed that GLP-1 Ra”, as you recommended, but with limitation to patients with
diabetes mellitus and COVID-19 as our study aim.
14. The meta-analysis by Patoulias et al. is not reference 31 but reference 34.
Response: Thank you for your observation, we rechecked the references and updated them.
15. Discussion: The discontinuation of SGLT2i has been argued in the context of acute
illness as a standard policy. See https://doi.org/10.1111/dom.14805.
Response: Thank you for your comment, we edited the phrase you indicated.
Round 2
Reviewer 2 Report
The authors have satisfactorily addressed my comments and revised the manuscript accordingly.